# Cerebral compliance assessment from intracranial pressure waveform analysis: Is a positional shift-related increase in intracranial pressure predictable?

Donatien Legé[1,2]*, Pierre-Henri Murgat[3], Russell Chabanne[5], Kevin Lagarde[6], Clément Magand[3], Jean-François Payen[6], Marion Prud'homme[2], Yoann Launey[4], Laurent Gergelé[7]

**1** DISC Department, FEMTO-ST, Université de Franche-Comté, Besançon, France, **2** Sophysa, Orsay, France, **3** Department of Anesthesia and Intensive Care, University Hospital of Saint-Etienne, Saint-Etienne, France, **4** Department of Anaesthesia, Critical Care Medicine and Perioperative Medicine, University Hospital of Rennes, Rennes, France, **5** Department of Anaesthesia, Critical Care Medicine and Perioperative Medicine, University Hospital of Clermont-Ferrand, Clermont-Ferrand, France, **6** Department of Anesthesia and Critical Care, University of Grenoble Alpes, CHU Grenoble Alpes, Grenoble, France, **7** Department of Intensive Care, Ramsay Générale de Santé, Hôpital privé de la Loire, Saint Etienne, France

☯ These authors contributed equally to this work.
* donatien.lege@femto-st.fr

**Data Availability Statement:** The raw data from the study cannot be shared publicly because data contains potentially identifying or sensitive patient

## Abstract

Real-time monitoring of intracranial pressure (ICP) is a routine part of neurocritical care in the management of brain injury. While mainly used to detect episodes of intracranial hypertension, the ICP signal is also indicative of the volume-pressure relationship within the cerebrospinal system, often referred to as intracranial compliance (ICC). Several ICP signal descriptors have been proposed in the literature as surrogates of ICC, but the possibilities of combining these are still unexplored. In the present study, a rapid ICC assessment consisting of a 30-degree postural shift was performed on a cohort of 54 brain-injured patients. 73 ICP signal features were calculated over the 20 minutes prior to the ICC test. After a selection step, different combinations of these features were provided as inputs to classification models. The goal was to predict the level of induced ICP elevation, which was categorized into three classes: less than 7 mmHg ("good ICC"), between 7 and 10 mmHg ("medium ICC"), and more than 10 mmHg ("poor ICC"). A logistic regression model fed with a combination of 5 ICP signal features discriminated the "poor ICC" class with an area under the receiving operator curve (AUROC) of 0.80 (95%-CI: [0.73—0.87]). The overall one-versus-one classification task was achieved with an averaged AUROC of 0.72 (95%-CI: [0.61—0.83]). Adding more features to the input set and/or using nonlinear machine learning algorithms did not significantly improve classification performance. This study highlights the potential value of analyzing the ICP signal independently to extract information about ICC status. At the patient's bedside, such univariate signal analysis could be implemented without dependence on a specific setup.

information Therefore, only the processed time series features are available in Supplementary Information. Original data are available from St-Etienne University Hospital Ethics Committee (contact via rgpd-dpd@chu-st-etienne.fr) for researchers who meet the criteria for access to confidential data.

**Funding:** The author(s) received no specific funding for this work.

**Competing interests:** Marion Prud'homme and Donatien Legé are employees of Sophysa Company. Laurent Gergelé has performed consulting work for Sophysa Company. This does not alter our adherence to PLOS ONE policies on sharing data and materials.

# Introduction

Measurement of intracranial pressure (ICP) is a standard aspect of monitoring in neuro-intensive care units. While the Brain Trauma Foundation recommends ICP measurement for cases of severe cranial trauma [1], both the Neurocritical Care Society and the European Society of Intensive Care Medicine suggest measuring ICP in patients showing clinical or radiological signs of intracranial hypertension [2]. However, despite the existence of these guidelines, the exact pathological upper threshold for ICP remains unclear. The Brain Trauma Foundation suggests using a threshold of 22 mmHg, whereas recent analysis of data from the European CENTER-TBI cohort shows a worsening prognosis at a threshold of $18mmHg \pm 2mmHg$ [3]. In any case, the effectiveness of ICP threshold-driven treatments is still debated. The BEST-TRIP randomized clinical trial in traumatic brain injury (TBI) patients found no significant difference in prognosis between the ICP-based treatment group and those treated based on clinical and imaging criteria [4]. Conversely, the observational SYNAPSE-ICU study across 146 centers worldwide showed that ICP monitoring was associated with more aggressive treatment, reduced mortality, and improved neurological prognosis [5]. The challenge of demonstrating a clear benefit from treating elevated ICP may be due to the fact that interventions are often applied too late, when structural brain damage has already occurred. Indeed, an elevated ICP is a de facto indication that the brain's compensation mechanisms have already been exceeded.

Beyond the mean ICP number itself, question arises about the opportunity of extracting more physiological information from ICP, with the underlying possibility of an early abnormality detection [6]. It is a complex mixture of periodic components, notably influenced by heart rate and respiratory rate [7]. For instance, the ICP signal morphology is known to be indicative of the cerebrospinal system ability to buffer volume variations without a substantial increase in mean ICP, also known as intracranial compliance (ICC) [8, 9]. As part of bedside multi-modal monitoring, real time ICC assessment can provide valuable information about acute brain damage pathophysiology [10], help anticipate adverse ICP elevations [11], and serve as an additional prognostic indicator [12]. However, the direct assessment of ICC is challenging due to the requirement of invasive procedures [13, 14] or advanced MRI imaging [15, 16].

Several methods for characterizing ICC through ICP signal processing have been proposed in the scientific literature, often with the intention of real-time calculation at the patient's bedside. These methods fall into two main categories. The first one focuses on the heartbeat-induced pulse shape [17, 18], often looking at the relative amplitudes of characteristic sub-peaks [19, 20]. The second category involves spectral analysis, either to study the higher frequency distribution [21] or to isolate respiratory and/or cardiac components [22, 23]. Despite the diversity of these approaches, only a few studies have correlated computerized ICP signal analysis with direct ICC measurements, and these have mainly been carried out in patients with chronic ICC disorders such as hydrocephalus [24, 25]. Although none of them have been integrated into routine clinical practice in the intensive care unit (ICU), several of these ICP-derived metrics show promise in helping clinicians identify patients with impaired ICC. Furthermore, to the best of our knowledge, no studies in the literature have yet explored the integration of multiple ICC surrogates calculated from the ICP signal alone for accurate ICC characterization.

# Materials and methods

This study examines how effectively a combination of these indices can be used to discriminate patients with impaired ICC. To this end, different machine learning (ML) models were fed

with various ICP signal features to predict patient responses to a 30-degree upper body tilt used as a quick ICC evaluation. This maneuver relies on the intricate relationship between ICP, ICC, and body positioning, which has continuously been investigated over the past four decades [26, 27]. In the four NICU involved in the present study, such a positional shift is routinely used by clinicians in NICU to quickly evaluate a patient's ICC and suitability for MRI or other imaging procedures. Using a reduced set of input ICP signal features, we seek to predict the positional shift-induced ICP elevation levels classified into three distinct categories: Inferior to 7 mmHg ("good ICC"), comprised between 7 and 10 mmHg ("medium ICC"), or superior to 10 mmHg ("poor ICC").

## Study population

Patients included to the study were admitted to the NICU of one of the four French University Hospitals involved in the study (Rennes, Clermont-Ferrand, Saint-Etienne, and Grenoble) between March 29, 2022, and March 22, 2024. Inclusion criteria were the following: Aged comprised between 18 and 80 years old, sedation under mechanical ventilation, and continuous high-frequency ICP monitoring (at least 100Hz for sampling). Exclusion criteria included major hemodynamic failure, adverse intracranial hypertension and decompressive craniectomy. Demographic data are summarized in Table 1.

Oral informed consent was obtained by the attending clinician and documented alongside the clinical data. If the patient was unable to provide consent, the clinician obtained the oral consent from their family members or relatives. Before obtaining consent, an informational notice, including contact information for withdrawal, was provided. The study was conducted in accordance with the Declaration of Helsinki and approved by the Ethics Committee of the University Hospital of Saint-Etienne (IRBN282022/CHUSTE, February 10, 2022).

## Protocol and data collection

The patients were treated according to the international guidelines for acute TBI management [28], following an ICP-based therapeutic algorithm [29]. The head was maintained at 30˚ throughout the treatment period, as systematically applied to all types of severe brain injury

**Table 1. Demographic data of the study cohort.**

| Variable | Observation |
| --- | --- |
| Number of patients | 54 |
| Age | 49.9 (17.8, 18–77) |
| Female | 16 / 30% |
| Male | 38 / 70% |
| TBI | 40 / 74% |
| aSAH | 9 / 17% |
| Other pathology | 5 / 9% |
| Initial GCS | 6.9 (3.2, 3–15) |
| Initial SAPSII | 48.8 (12.3, 23–71) |
| ICC tests performed | 2 (1.8, 1–9) |
| Length of stay (days) | 25.2 (13.3, 4–53) |
| 28-day mortality | 10 / 18.5% |

TBI: Traumatic brain injury, aSAH: Aneurysmal subarachnoid haemorrhage, GCS: Glasgow coma scale, SAPSII: Simplified Acute Physiology Score, ICC: Intracranial compliance. When applicable, numbers are expressed in the format: mean (std, minimum—maximum).

with ICP monitoring in all NICUs involved in the study. For each patient, a 100 Hz-resolution ICP signal was acquired with an intraparenchymal sensor (Pressio, Sophysa, Orsay, France). An ICC test consisted of temporarily lowering the patient's upper body down to 0˚ until ICP signal stabilization and for at least two minutes. The associated ICP elevation was calculated as the difference between the median ICP measured on the maneuver-induced ICP plateau and the median ICP measured over the previous ten minutes. The clinician also reported the following variables:

- Natremia measured within the 12 hours preceding the positional shift.

- Body temperature measured by the means of vesical probe or tympanic measurements.

- Mean arterial pressure (MAP) measured by invasive arterial catheter at heart level, immediately before the positional shift.

- Cerebral perfusion pressure (CPP), calculated as following: CPP = MAP—ICP.

Most patients underwent multiple ICC tests as the investigator observed changes in their clinical status. After exporting the data, ICC tests that met the following criteria were included in the study:

- At least twenty minutes of good quality 100Hz-resolution ICP signal prior to the ICC test were available.

- It was possible to calculate the ICP elevation caused by the ICC test (since signals could be confused by coughing, acute ICP elevation before the ICC test, etc.).

Initially, 111 ICC tests were performed on 55 patients. In view of the above-mentioned criteria, only 108 tests spread over 54 patients were considered for data analysis.

## ICP signal analysis

For each patient, the 20-minute ICP signal recorded immediately before the positional shift was characterized by calculating 73 different features. The only preprocessing algorithm applied to the raw ICP signal was a Butterworth fourth-order low-pass filter whose threshold was set to 20 Hz. After a selection step, different combinations of these features were provided as inputs to classification models. The overall classification process is summarized in Fig 1.

The analysis described below was implemented with Python 3.11. For reproducibility purposes, the different scripts were included into a pipeline managed with Snakemake 8.9.0. [30] The experiments were run on a Windows 10 machine powered by WSL2 Ubuntu 20.04.5, equipped with a 12th Gen Intel(R) Core(TM) i7–12850HX 2.10 GHz 16 CPU, an Nvidia RTX A3000 12GB Laptop GPU, and 16 GB of RAM.

**ICP signal descriptors.**    Numerous ICC indicators based on the ICP signal morphology have been proposed in the literature over the past thirty years [31, 32]. We selected a few of them to be used as inputs to our classification models, in addition to the raw signal itself. These are the following:

- **ICP Pulse Amplitude (AMP)**. It corresponds to the amplitude of each cardiac-induced pulse of the ICP signal. Assuming that the part of the heart stroke volume transmitted to the brain remains constant, AMP can be interpreted as a consequence of the pressure-volume relationship in the cerebrospinal compartment. Of all the ICP signal-based ICC surrogates, AMP is probably the most studied in the literature [33–35]. In the present study, the modified Scholkmann algorithm [36] was used to isolate each single ICP pulse and then compute peak-to-nadir amplitudes.

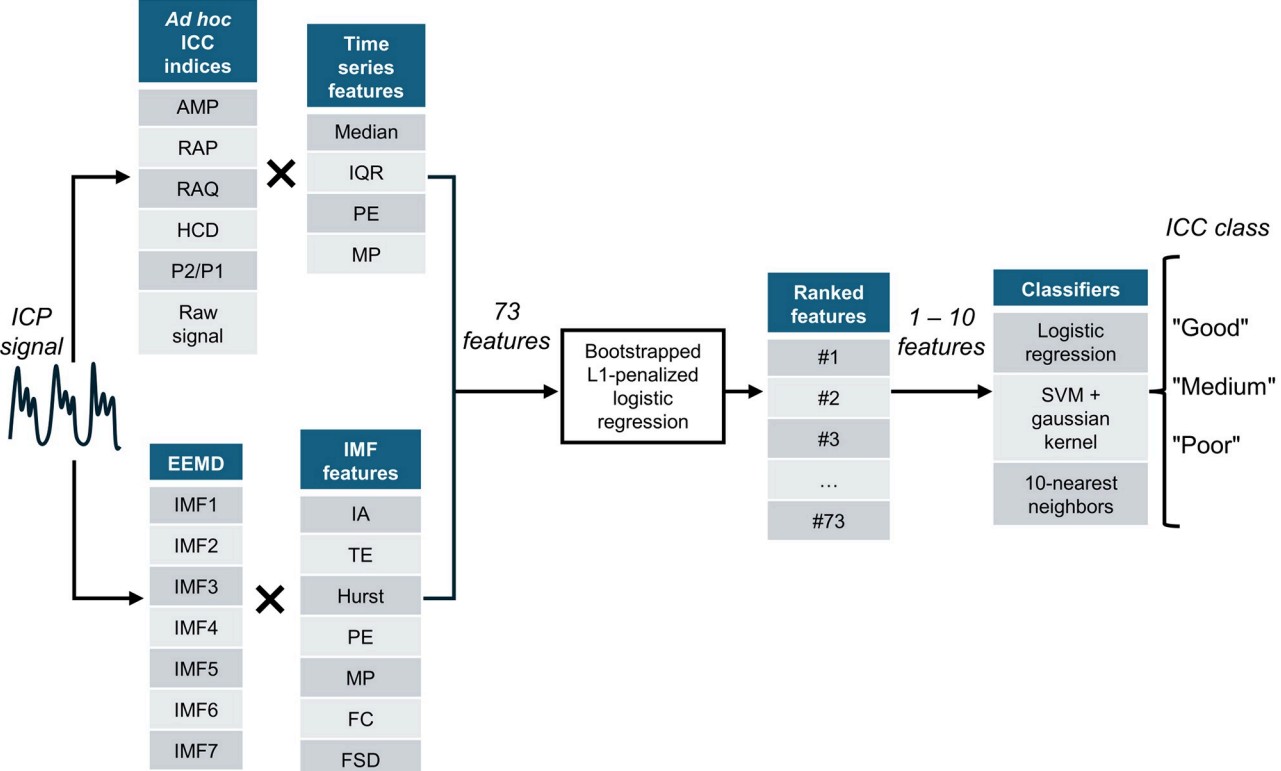

**Fig 1. Feature extraction and selection process.** ICC: Intracranial compliance, AMP: ICP Pulse amplitude, RAP: RAP index, RAQ: RAQ index, HCD: Heart cycle duration, P2/P1: P2/P1 ratio, EEMD: Ensemble empirical mode decomposition, IQR: Interquartile range, PE: Permutation entropy, MP: Missing patterns, IA: Instantaneous amplitude, TE: Teager energy, FC: Frequency centroid, FSD: Frequency standard deviation, SVM: Support vector machine.

- **P2/P1 ratio**. The shape of the heartbeat-induced pulses is known to vary with ICC. It can be characterized by the relative heights of three sub-peaks, namely P1, P2, and P3, which usually appear successively on each pulse. While P1 is caused by the systolic volume entering the brain, P2 is most often considered to be a reflection wave that coincides with a maximum volume into the cerebral arteries [37]. As ICC decreases, P2 becomes more prominent until the pulse takes a triangular shape centered around it. Therefore, the ratio of the relative amplitudes of P2 and P1 (designated as the P2/P1 ratio) can be used as an indicator of ICC [24]. In the present study, the automated P2/P1 ratio calculation was performed using the framework previously developed by the authors [38].

- **RAP Index**. The RAP index is defined as the moving Pearson correlation coefficient between mean ICP and AMP. Its relationship with ICC has been broadly studied since its definition in the late 1980s [12, 39, 40], despite a few shortcomings regarding its robustness to baseline errors [41]. In the present study, the RAP index was calculated over a 40-pulse sliding window detected with the modified Scholkmann algorithm, hence without leaving the time domain.

- **RAQ Index**. The RAQ index describes how much the ICP signal respiratory component affects AMP. It is formally defined as the ratio of the respiratory component amplitude to the respiratory-induced changes in AMP. Its authors conceived it to be as robust as possible to spurious baseline changes (manipulations by medical staff, for instance) [42]. The RAQ

index is expected to decrease with increasing ICC until a breakpoint, beyond which it tends to infinity.

- **Heart Cycle Duration (HCD)**. After having detected the heartbeat-induced pulses by the means of the modified Scholkmann algorithm, the onset-to-onset duration is extracted as a surrogate for heart rate variability (HRV). While not specifically studied in the context of ICC, it is broadly admitted that a high HRV can reflect autonomic nervous system disorders [43], which is also a possible cause for the presence of ICP slow waves [44].

- **Intrinsic Mode Functions (IMFs)**. Empirical Mode Decomposition (EMD) is a signal decomposition technique developed by Huang *et al.* to circumvent the theoretical assumptions of Fourier analysis, in particular the assumption of stationarity [45]. The input signal is decomposed into intrinsic mode functions (IMFs), which possess mathematical properties that enable the precise definition of instantaneous phase (IP), frequency (IF) and amplitude (IA). EMD is a powerful general-purpose tool that has notably been used to detect artifacts in the ICP signal [46] and even to forecast it [47]. As its original sifting algorithm is subject to the mode mixing problem [48], IMFs are here extracted with a noised-assisted version called ensemble EEMD (EEMD) [49] implemented with Python library EMD [50]. After discarding the 0-th IMF (it corresponds to the artificial noise added to the signal to stabilize the decomposition algorithm), different features are extracted from the first seven IMFs. These IMFs cover a patient-specific frequency range from approximately 0.02 Hz to 10 Hz. An example of such a decomposition on a 60-second ICP signal is presented Fig 2.

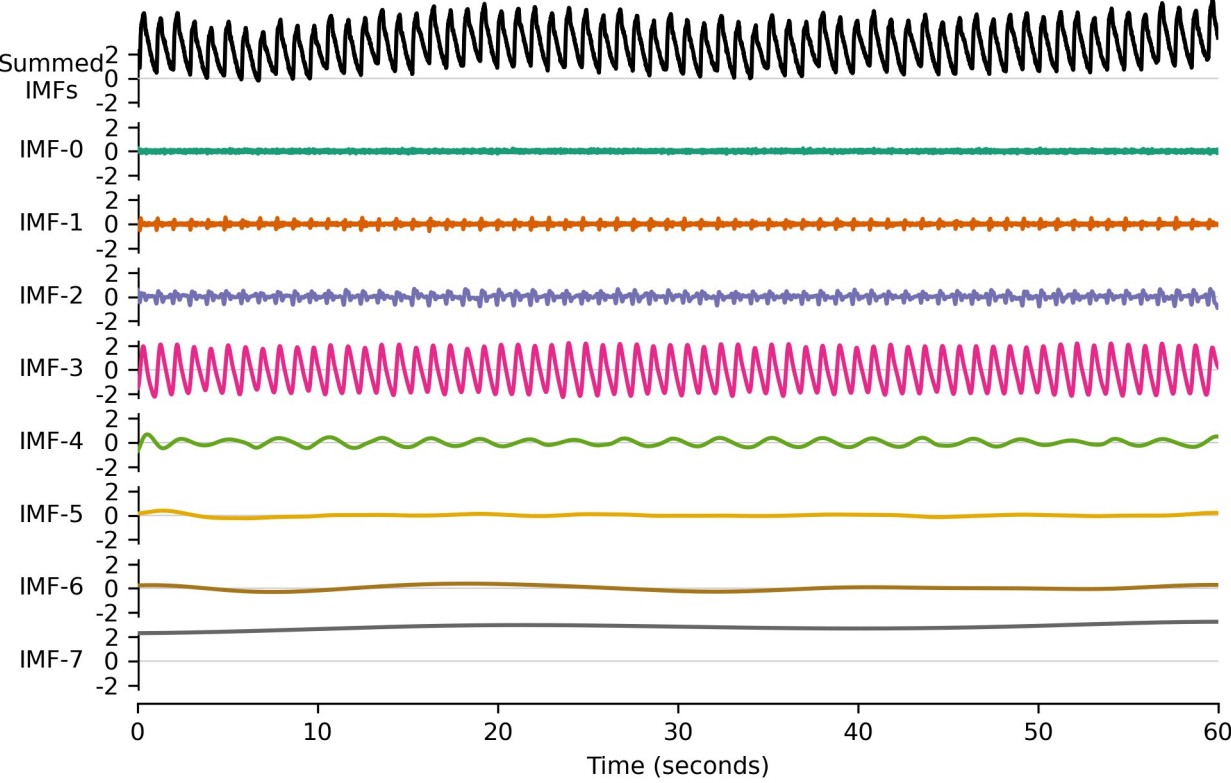

**Fig 2. Seven intrinsic mode functions (IMF) iteratively extracted from a 60-second intracranial pressure signal with ensemble empirical mode decomposition.**

**Feature extraction process.** Five ICC indices previously described (AMP, P2/P1 Ratio, RAQ, RAP, HCD) are calculated over a 20-minute period prior to the ICC test. The EEMD algorithm is then applied to the same monitoring section in order to decompose the signal into seven IMFs. As a result, thirteen times series are extracted from the ICP signal: five ICC indices, seven IMFs, and the raw signal itself. Whereas the same features are extracted from the ICC indices and from the raw ICP signal, specific calculations are performed on the seven IMFs. This distinction is made to take into account for mathematical properties imposed by the EEMD algorithm. For instance, calculating a mean or a median would not make any sense since an IMF is by definition centered around 0. As a result, 73 features are extracted from each 20-minute monitoring period. The final design matrix is available in S1 Data.

The following features are calculated for each of the time series corresponding to the evolution of AMP, P2/P1 Ratio, RAQ, RAP, HCD, as well as for the raw ICP signal itself:

- **Median**.

- **Interquartile range (IQR)**. It corresponds to the difference between the third and the first quartiles of the time series values.

- **Permutation entropy (PE)**. Meant to characterize the complexity of a time series, PE is an instance of the Shanon discrete information entropies, all of them being defined for a time series $s$ as following (admitting that $0log(0) = 0$):

$$Entropy(s) = -\sum_{i\in I} p_i log(p_i) \tag{1}$$

In the case of PE, the set $I$ corresponds to all the ordinal patterns that can be generated with a $n$-point sliding window, and $p_i$ to the empirical frequency associated with pattern $i$. For instance, with $n = 3$, the sequence (1.5, 0, 2.5) read by the sliding window would be mapped to the pattern (2, 1, 3). Intuitively, PE tends to 0 as the time series approaches a straight line, and tends to $log(n!)$ as the time series approaches a white noise. In the present study, $PE$ was calculated with $n = 6$ using the Python library Ordpy [51].

- **Missing Patterns (MP)**. During the PE calculation process described above, the ratio of unobserved patterns out of the theoretically possible 6! = 720 ones is extracted as a signal descriptor. As demonstrated on a porcine model, MP tends to increase with intracranial hypertension and cerebral circulation arrest [52].

The following features are calculated over the seven extracted IMFs:

- **Instantaneous Amplitude (IA)**. An IMF $x$ can be prolonged to an analytical signal $\sigma$ such that $\sigma(t) = x(t) + j\hat{x}(t) = IA(t)e^{j\theta(t)}$, where $j$ denotes the imaginary unit and $\hat{x}$ the Hilbert transform of $x$. In the exponential form of this complex-valued function, the instantaneous phase (IP) corresponds to the argument $\theta(t)$, the instantaneous frequency (IF) to its derivative $\frac{d\theta(t)}{dt}$, and the instantaneous amplitude (IA) to the module $IA(t)$.

- **Teager Energy (TE)**. For a time series $x$ of length $N$, It is defined as following for a time series: $TE(x) = log(\sum_{t=2}^{N-1} x(t)^2 - x(t-1) * x(t+1))$

- **Hurst Exponent ($H$)**. It is an estimator of the time series self-similarity. The classical interpretation distinguishes between three cases. A $H$ less than 0.5 means that if the time series has been above its mean, it is more likely to fall in the future, and vice versa. A $H$ superior to 0.5 implies some level of trend-following behavior, i.e., a high (low) value is likely to be

followed by a high (low) value. When $H$ is close to 0.5, the time series is close to a random noise.

- **Permutation Entropy**, as described for general time series.

- **Missing Patterns (MP)**, as described for general time series.

- **Frequency Centroid (FC)**. It corresponds to the average of IFs weighted by their associated IAs, i.e.

$$FC(x) = \frac{\sum_{i=1}^{N} IF(i) * IA(i)}{\sum_{i=1}^{N} IA(i)} \tag{2}$$

- Frequency Standard Deviation (FSD). It is calculated as the standard deviation of IFs weighted by their associated IAs such that

$$FSD(x) = \sqrt{\frac{\sum_{i=1}^{N} IA(i) * (IF(i) - FC(x))^2}{(N-1)\sum_{i=1}^{N} IA(i)}} \tag{3}$$

**Feature selection and classification process.** The goal of the classification task is to predict a patient's response level to the postural shift, i.e. whether the ICP elevation is going to be inferior to 7 mmHg ("good ICC"), comprised between 7 and 10 mmHg ("medium ICC") or superior to mmHg ("poor ICC"). In the rest of the article, $X_y$ denotes a feature $y$ calculated from a time series $X$.

The feature selection procedure relies on a BoLasso algorithm [53] adapted for a classification task. This method consists of ranking the features based on a l1-penalized logistic multinomial regression (l1-LR). More formally, the regression model aims to estimate the coefficient matrix $B$ such that the estimate probability $\hat{p}_k(X_i)$ of the $i$-th postural shift belonging to the $k$-th class is the following:

$$\hat{p}_k(X_i) = \frac{e^{X_i B_k + B_{0,k}}}{\sum_l^K e^{X_i B_l + B_{0,l}}} \tag{4}$$

$B$ counts $N + 1$ rows and $K$ columns, where $N$ denotes the number of features (73 in our case) and $K$ the number of classes (3 in our case). The 0-th row corresponds to the intercept. The SAGA solver in the Python library Scikit-learn [54] is used to find a matrix B that minimizes:

$$-C\sum_{i=1}^{N}\sum_{k=1}^{K} \mathbb{1}_k(y_i) log(\hat{p}_k(X_i)) + \|B\|_{1,1} \tag{5}$$

where C is the regularization strength parameter left to the user's choice, $y_i$ denotes the true class of the $_i$-th test, $\mathbb{1}_k(y_i)$ is equal to 1 if $y_i = k$, 0 otherwise, and $\|B\|_{1,1} = \sum_{i=1}^{N}\sum_{j=1}^{K}\|B_{i,j}\|$. The regularization strength controls how sparse the $B$ matrix is going to be. The sparseness of the coefficient matrix is a crucial property on which the ranking process is entirely based. Considering the design matrix $X$ and the true class vector $Y$, the following procedure is applied:

1. Feature are standardized to ignore scale disparities. The matrix $X^*$ is obtained as follows: $X^* = \frac{X - \bar{X}}{\sigma(X)}$ where $\bar{X}$ denotes the mean of X and $\sigma(X)$ its standard deviation.

2. The regularization strength $C$ is determined using a 5-fold cross validation (CV). To do so, a l1-LR model is fitted to predict $Y$ from $X^*$ with different values of $C$. The value of $C$ that leads to the best average accuracy (i.e., proportion of good predictions) over the 5 folds is selected for the following step and denoted $C^*$.

3. 1000 bootstrap samples are randomly drawn from $X^*$. For each of them, a coefficient matrix $B_l$ is calculated from a $C^*$-penalized l1-LR model.

4. A relevance score is assigned to each of the 73 features. It corresponds to the proportion of matrices $B_l$ whose $i$-th row contains at least one non-zero coefficient.

Afterwards, three classification models are fed with the $n$ best ranked features, with $n$ ranging from 1 to 10. The optimal number of input features is chosen to maximize areas under the receiver operating curves (AUROC) of three classification algorithms:

- **Non-penalized logistic regression (LR)**. It corresponds to the same model as the one used for feature selection, but without the regularization term $\|B\|_{1,1}$

- **K-Nearest neighbors classifier**. Each new individual is put into the most frequent class among its $k$ nearest neighbors, weighted by their respective euclidean distance. In the present study, we set $k = 10$.

- **Support vector machine (SVM)**. This algorithm seeks to find the hyperplane that maximizes the margins between input classes. Since not all problems are linearly separable, SVMs are generally associated with a kernel function that basically maps initial coordinates into a more appropriate space for the problem instance. In the present study, we used a Gaussian kernel, which is among the most common ones.

To compare classification performances, multiple AUROC are calculated over 40 iterations of a 5-fold CV. As ROC curve is only defined for binary classification problems, two aggregation methods are considered: One-versus-one and one-versus-rest. In the first case, a ROC curve is calculated for each pair of classes (for instance, "poor ICC" versus "good ICC"). In the second case, a ROC curve is calculated for each isolated class (for instance, "poor ICC" versus "good ICC" or "medium ICC").

## Results

### Patient responses

The positional shift increased ICP by an average of 8.7 mmHg (std = 3.4 mmHg, min = 2.8 mmHg, max = 20.1 mmHg). However, the change in position had no clear effect on ICP morphology. As seen in Table 2, no significant variations in AMP nor in the P2/P1 ratio were observed during the maneuver (Wilcoxon signed-rank test $p$-values are equal to 0.07 and 0.94, respectively).

**Table 2. Positional shift-induced ICP signal variations.**

|  | 30˚ | 0˚ | 0˚ − 30˚ | *p*-value |
|---|---|---|---|---|
| ICP (mmHg) | 7.0 (5.5) | 15.7 (6.7) | 8.7 (3.4) | $< 10^{-4}$ |
| AMP (mmHg) | 7.2 (3.4) | 7.3 (4.0) | 0.1 (2.2) | 0.07 |
| P2/P1 Ratio | 1.26 (0.57) | 1.34 (0.71) | −0.08 (0.68) | 0.94 |

Numbers are expressed in the format: Mean (std). ICP: Intracranial pressure, AMP: ICP Pulse Amplitude. For each measurement, a Wilcoxon signed-rank test was performed between the measurements at 30˚ and 0˚.

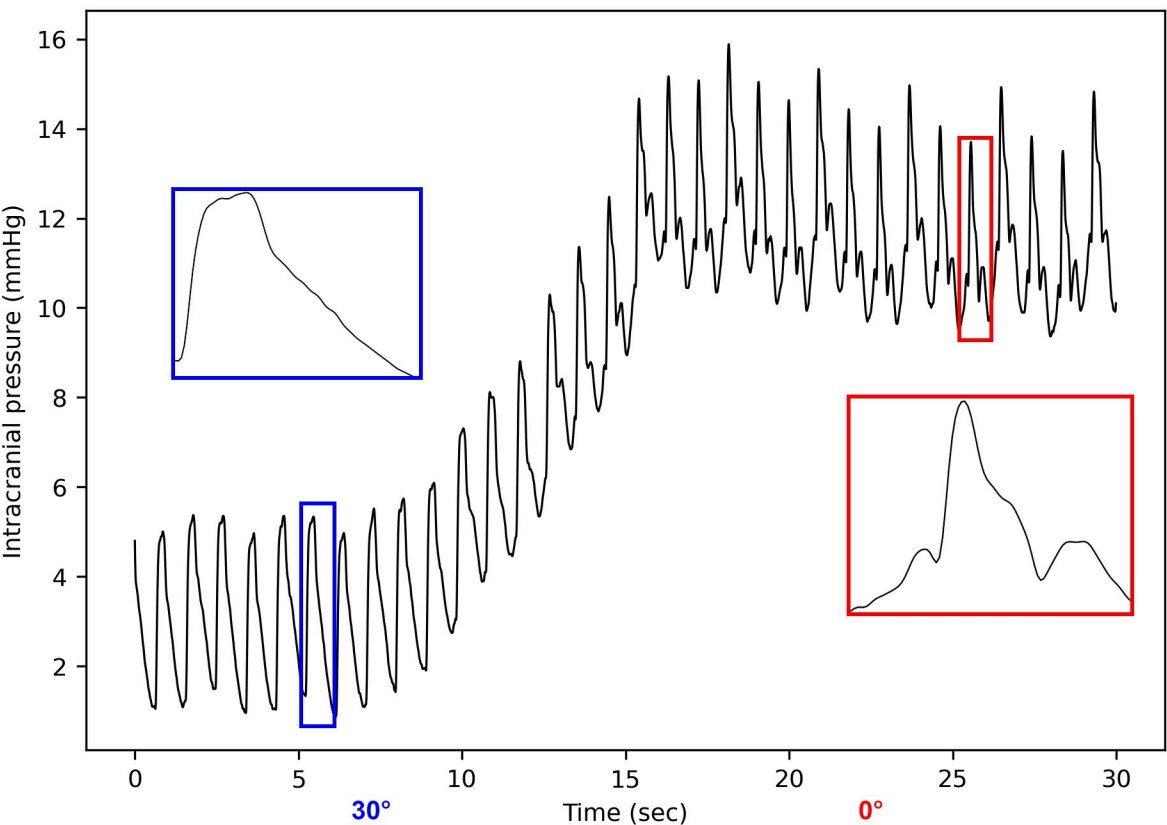

**Fig 3. Example of a strong intracranial pressure morphology change induced by a 30˚ postural shift.**

The postural shift caused highly variable changes in the ICP pulse morphology, which were difficult to quantify with the sole P2/P1 ratio. Whereas the overall pulse shape remained unchanged in most of the cases, the maneuver sometimes led to changes as significant as shown in Fig 3.

## Effects of clinical and biological variables on patient responses

Relationships between ICP elevation and several clinical or biological variables were also investigated. Kruskall-Wallis and Fisher's exact tests of independence were performed to identify potential links between patient-specific variables and ICC classes. Results are presented in Table 3.

None of the variables studied seemed to significantly affect the response to the positional shift. In particular, arterial blood pressure-related variables (i.e., MAP and CPP measured before the positional shift) did not differ between ICC classes ($p$-values were equal to 0.15 and 0.79, respectively). Furthermore, ventilation mode and CSF drainage devices did not appear to impact the responses to the positional shift.

## ICP signal harmonic structure

EEMD was performed to decompose the ICP signals into simpler waveforms called IMFs. For a more accurate description of the harmonic structure of the ICP signal, basic descriptors of the frequency distribution were calculated for each of the extracted IMF. The

**Table 3. Relationships between intracranial compliance classes and several patient-specific variables.**

| Variable | good ICC | Medium ICC | Poor ICC | p-value |
|---|---|---|---|---|
| Age | 57.9 (14.8) | 51.7 (18.8) | 49.5 (17.7) | 0.24[k] |
| Sex (% F) | 30.3 | 24.4 | 20.5 | 0.64[f] |
| Initial GCS | 6.5 (3.0) | 5.8 (3.1) | 6.5 (3.3) | 0.20[k] |
| Initial SAPSII | 50.8 (12.7) | 48.1 (13.3) | 53.2 (9.9) | 0.29[k] |
| EVD or ELD (%) | 36.3 | 39.0 | 20.5 | 0.21[f] |
| Ventilation mode (% ACV) | 54 | 64 | 73 | 0.50[f] |
| ICP (mmHg) | 6.98 (4.84) | 6.93 (5.86) | 7.67(4.91) | 0.77[k] |
| MAP (mmHg) | 80.6 (9.8) | 82.9 (14.2) | 84 (12.2) | 0.15[k] |
| CPP (mmHg) | 74.3 (10.7) | 76.1 (12.5) | 76.4 (13.2) | 0.79[k] |
| Temperature (˚C) | 37.14 (0.72) | 37.93 (0.91) | 36.73 (1.21) | 0.39[k] |
| Natremia (mEq/L) | 143.7 (4.5) | 141.3 (5.0) | 143.3 (3.8) | 0.12[k] |

When applicable, numbers are expressed in the format Mean (std). GCS: Glasgow Coma Score, SAPSII: Simplified Acute Physiology Score, EVD: External Ventricual Deviation, ELD: External Lumbar Deviation, ACV: Assist-Control Ventilation.

[k]: p-value of a Kruskall-Wallis test.

[f]: p-value of a Fisher's exact independence test.

frequency centroid corresponds to the mean of the frequencies weighted by their amplitudes, while the FSD corresponds to the standard deviation of these amplitude-weighted frequencies.

Each IMF covers an adaptive frequency range specific to each signal. The distribution of each IMF centroid over the entire dataset is presented in Fig 4. IMFs #1 to #3 capture information associated with heartbeat-induced pulses, with frequency centroids located above 1 Hz. Respiratory components and slower waves are contained in IMFs #4 to #7.

## Feature selection process

Calculated features were ranked based on the non-null coefficients of a bootstrapped l1-LR regression. Afterwards, best ranked features were added to the classification algorithm input set until the associated AUROC stopped increasing. Based on the results of a preliminary 5-fold CV procedure across all the dataset, the regularization strength was set to 0.2. The ten best ranked features are presented in Table 4. These features obtained at least 54.7% of rows with at least one non-null coefficient in the bootstrapped-lasso procedure, whereas the mean percentage was of 29.5% (std = 21%) over the entire feature set. Interestingly, only two of them differed significantly between ICC classes. The "good ICC" class showed comparatively lower RAP_entropy values (p-value = 0.02), whereas the "poor ICC" class was associated with a higher amplitude of IMF6, i.e. with the presence of slow waves (p-value = 0.01).

Notably, we did not have to face multicollinearity issues among the ten best ranked features. Whereas the strongest one-to-one Spearman correlation was of -0.60 between features #5 and #6, the condition index calculated over the standardized features appeared to be as low as 3.01. The associated correlation matrix is presented in Fig 5. Interestingly, even if two of the five highest ranked features characterize the IMF2 morphology, the Spearman correlation coefficient between both is not significantly different from 0 (correlation = 0.13, p-value = 0.17).

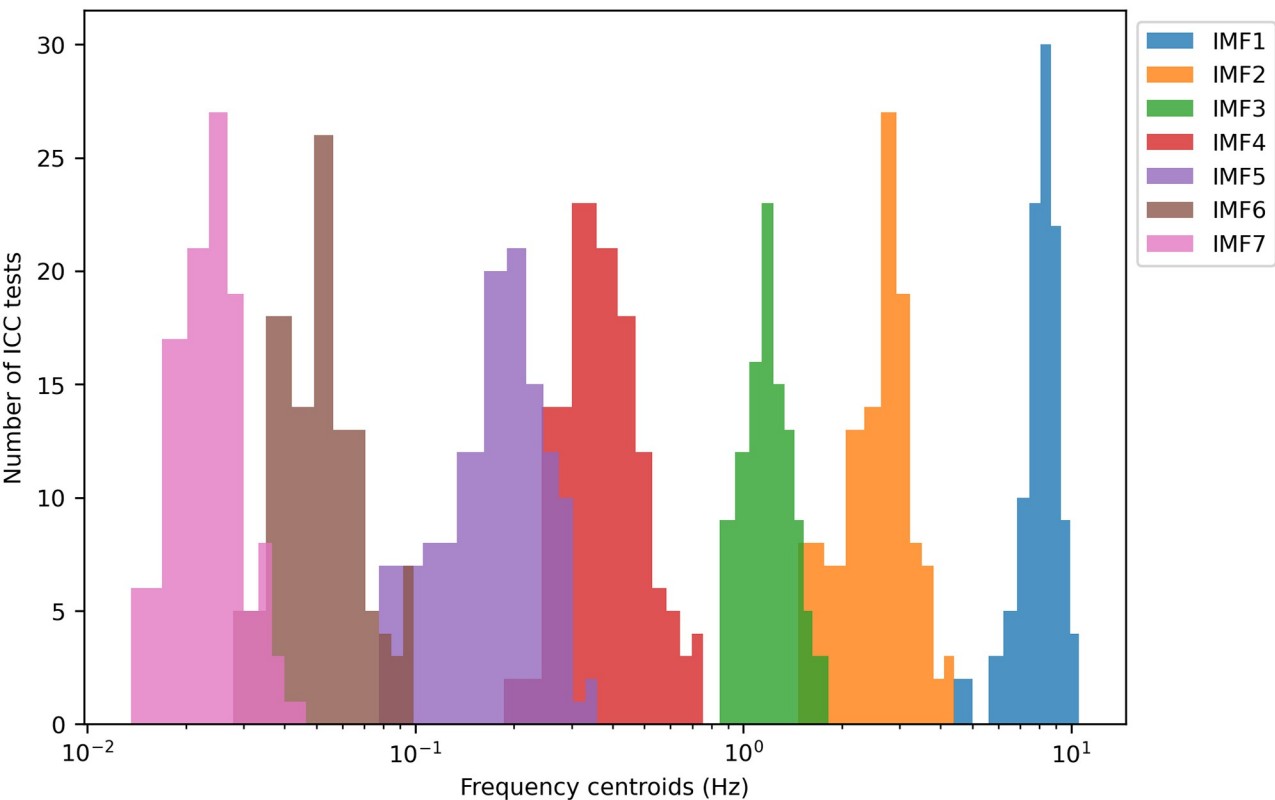

**Fig 4. Intrinsic mode functions (IMF) frequency centroids calculated from the intracranial pressure signals monitored in the 20 minutes preceding the positional shift.** Each histogram corresponds to an IMF centroid distribution over the whole dataset.

### Effects of ventilation and CSF derivation on ICP signal features

Mechanical properties of the cerebrospinal system can be modified due to controlled mechanical ventilation or external CSF derivation devices. Therefore, Mann-Whitney U tests were conducted to measure the effects of assist-controlled ventilation (ACV) and external ventricular

**Table 4. The ten best ranked ICP signal features.**

| Feature | Score (%) | Good ICC | Medium ICC | Poor ICC | *p*-value |
|---|---|---|---|---|---|
| IMF4_hurst | 89.7 | 0.73 (0.24) | 0.69 (0.25) | 0.83 (0.14) | 0.31 |
| IMF6_amplitude | 78.1 | 0.48 (0.33) | 0.45 (0.29) | 0.70 (0.45) | 0.01 |
| AMP_median | 77.6 | 7.5 (2.9) | 6.22 (2.2) | 8.83 (4.9) | 0.08 |
| IMF2_centroid | 73.6 | 2.55 (0.53) | 2.86 (0.58) | 2.80 (0.67) | 0.11 |
| IMF2_hurst | 67.6 | 0.27(0.09) | 0.24 (0.06) | 0.24 (0.07) | 0.21 |
| IMF4_centroid | 62.2 | 0.38 (0.01) | 0.44 (0.14) | 0.43 (0.11) | 0.17 |
| RAP_entropy | 62.0 | 0.79 (0.07) | 0.83 (0.07) | 0.81 (0.08) | 0.02 |
| IMF7_hurst | 57.8 | 0.972 (0.001) | 0.973 (0.001) | 0.972 (0.001) | 0.06 |
| AMP_entropy | 55.7 | 0.82 (0.09) | 0.79 (0.10) | 0.78 (0.10) | 0.34 |
| IMF7_fsd | 54.7 | 0.012 (0.004) | 0.012 (0.003) | 0.011 (0.003) | 0.14 |

The displayed *p*-value corresponds to a Kruskall-Wallis test between the three groups. The score corresponds to the percentage of non-null coefficient over 1000 iterations of a bootstrapped lasso regression. ICC: Intracranial compliance. IMF: Intrinsic Mode Function. AMP: ICP signal amplitude. RAP: RAP index. FSD: Frequency standard deviation

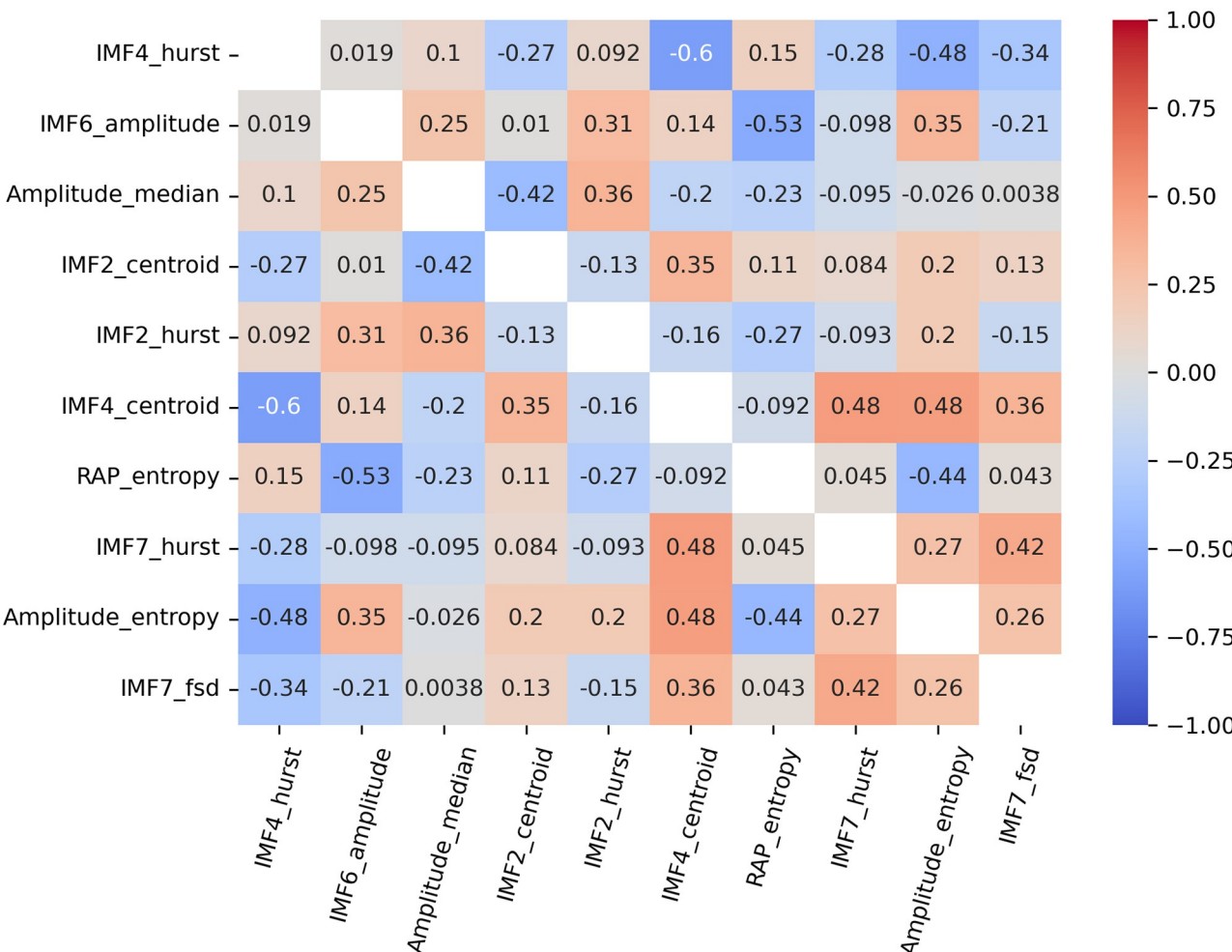

**Fig 5. Correlation matrix over the ten best ranked features.** IMF: Intrinsic Mode Function. Hurst: Hurst Exponent. FSD: Frequency Standard Deviation.

or lumbar CSF drainage (EVD or ELD, respectively) on the ten highest-ranked features. Results are presented in Table 5.

Mechanical ventilation seemed to have little influence on ICP morphology. None of the statistical test null hypotheses could be rejected. Variables explicitly linked to the respiratory wave time scale (i.e., those that characterize the IMF4 morphology) did not exhibit significant differences in patients placed or not under assist-control ventilation (ACV). In contrast, ICP signal from patients equipped with a CSF drainage device were changed in low frequency ranges. Use of CSF drainage appeared to cause a two-fold decrease in the amplitude of the IMF6, whose frequency centroid is located around 0.05 Hz over the whole dataset. Other features, especially those associated with high frequencies, remained comparable between patients equipped or not with a CSF derivation device.

## Effects of biological variables on ICP signal features

Spearman correlations were calculated between biological variables and the ten best-ranked features. Results are summarized in Fig 6.

**Table 5. Relationships between the ten best-ranked ICP signal features and categorical patient-specific variables.**

| Feature | CSF drainage | | | Ventilation | | |
|---|---|---|---|---|---|---|
| | With | Without | *p*-value | SAV | ACV | *p*-value |
| IMF4_hurst | 0.77 (0.20) | 0.69 (0.27) | 0.29 | 0.67 (0.25) | 0.75 (0.22) | 0.08 |
| IMF6_amplitude | 0.62 (0.40) | 0.34 (0.26) | $< 10^{-4}$ | 0.52 (0.27) | 0.54 (0.40) | 0.71 |
| AMP_median | 7.67 (3.9) | 6.95 (3.0) | 0.50 | 8.81 (3.94) | 7.75 (3.7) | 0.28 |
| IMF2_centroid | 2.76 (0.59) | 2.73 (0.64) | 0.78 | 2.59 (0.48) | 2.76 (0.62) | 0.24 |
| IMF2_hurst | 0.26 (0.08) | 0.24 (0.07) | 0.24 | 0.26 (0.06) | 0.25 (0.07) | 0.34 |
| IMF4_centroid | 0.42 (0.12) | 0.41 (0.12) | 0.64 | 0.42 (0.09) | 0.41 (0.12) | 0.58 |
| RAP_entropy | 0.80 (0.07) | 0.82 (0.07) | 0.16 | 0.78 (0.08) | 0.82 (0.07) | 0.09 |
| IMF7_hurst | 0.97 (0.001) | 0.97 (0.001) | 0.24 | 0.97 (0.001) | 0.97 (0.001) | 0.13 |
| AMP_entropy | 0.80 (0.09) | 0.79 (0.13) | 0.87 | 0.82 (0.08) | 0.77 (0.09) | 0.15 |
| IMF7_fsd | 0.01 (0.003) | 0.01 (0.003) | 0.09 | 0.01 (0.003) | 0.01 (0.002) | 0.72 |

*p*-values correspond to a Mann-Whitney U test between both modalities of each variable. EVD: External ventricual derivation, ELD: External lumbar derivation, SAV: Spontaneous assisted ventilation, ACV: Assist-control ventilation.

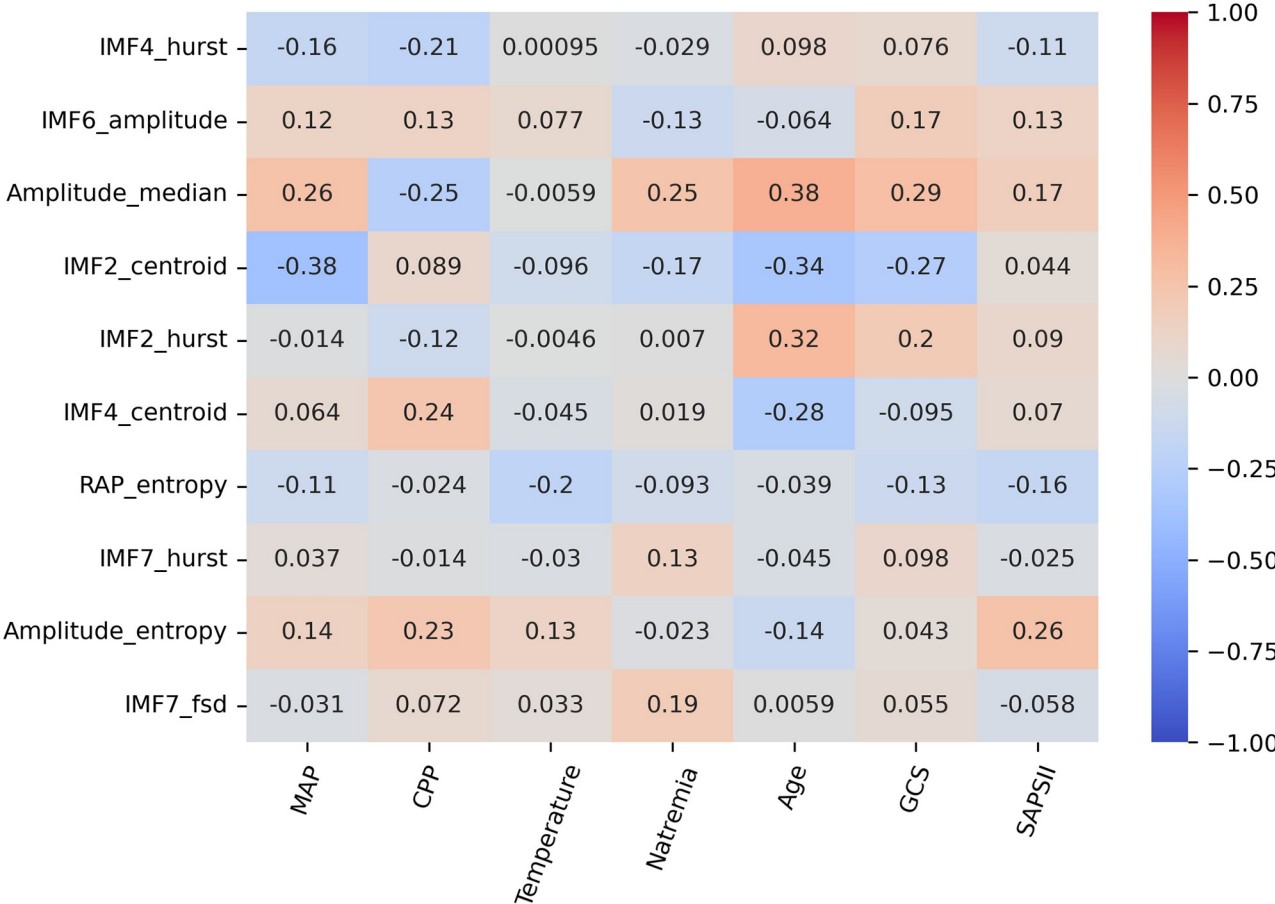

**Fig 6. Spearman correlation coefficients between the ten highest ranked ICP signal features and five biological variables.** IMF: Intrinsic Mode Function, FSD: Frequency Standard Deviation, MAP: Mean Arterial Pressure, CPP: Cerebral Perfusion Pressure.

Patient age showed significant correlations with several ICP characteristics. In particular, older patients exhibited ICP pulse characteristics that are known to reflect impaired ICC. Indeed, older patients tended to be associated with ICP pulses of highest amplitude (Spearman correlation = 0.38) and of more triangular shapes as described by the IMF2's Hurst exponent (Spearman correlation = 0.32, see figures in S1 and S2 Figs for visual interpretation). MAP was correlated with lower frequency centroids of IMF2 (Spearman correlation = -0.34). Only moderate correlations with ICP signal morphology were found for the other variables.

## Classification algorithms

Model performances were assessed by computing AUROCs averaged over 40 iterations of a 5-fold CV. As shown in Fig 7 in the case of a binary sub-task, providing $n = 5$ features to the classification algorithms appeared to be the best compromise between performance and complexity. The detailed AUROCs obtained for $n = 5$ are presented in Table 6. Although all of the three classification models exhibited very comparable performance, a simple logistic regression performed slightly better on average with a one-versus-one AUROC of 0.72 ± 0.11 (95%-CI width). Increasing the input set or using nonlinear classification model did not improve the calculated performance. Among all the possible binary classification tasks, discriminating the "poor ICC" class was achieved with the highest AUROC (0.80 ± 0.11).

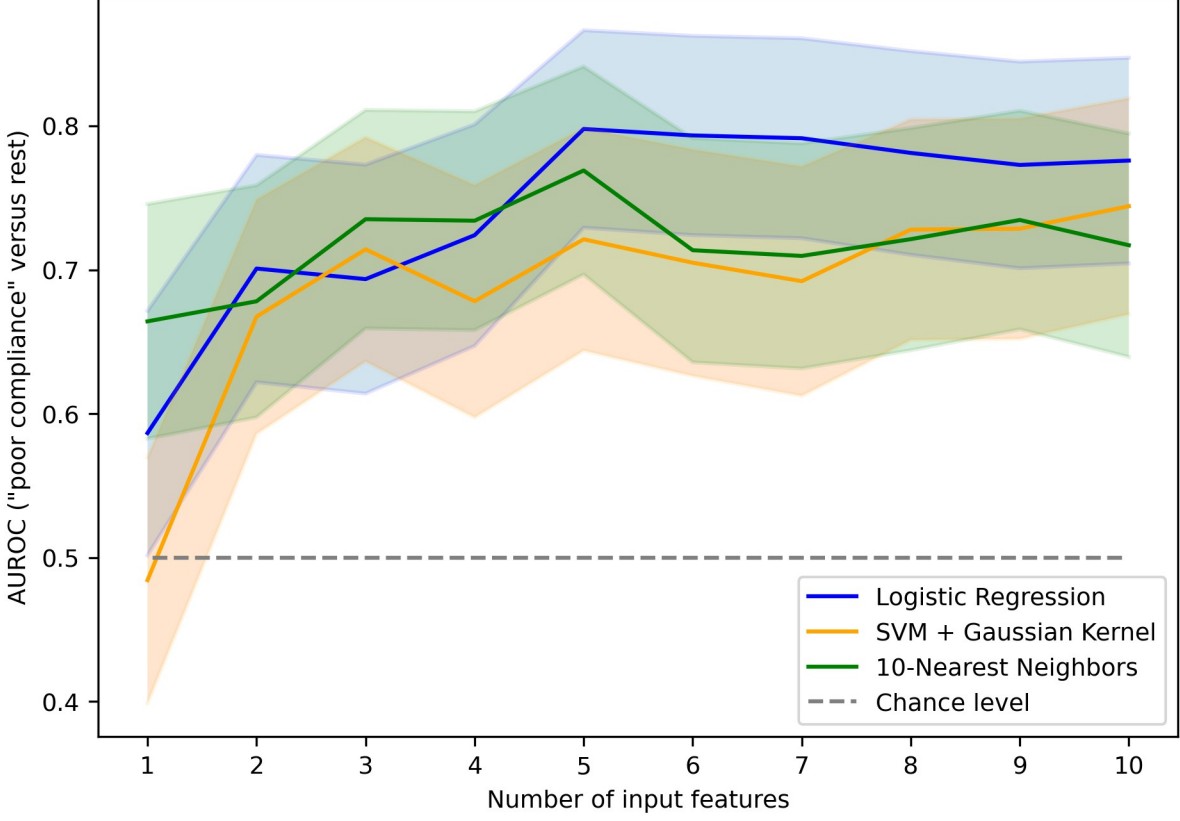

**Fig 7. Area under the receiver operating curve of the classifiers as a function of the number of input features.** Models were evaluated here on their ability to discriminate patients with impaired intracranial compliance ("poor compliance" class). SVM: Support Vector Machine. Shaded areas correspond to a 95% confidence interval.

**Table 6. Compared performances of the classifiers trained with the 5-feature input set.**

| Algorithm | AUROC (one vs one) | AUROC (poor vs rest) | AUROC (good vs rest) |
|---|---|---|---|
| LR | 0.72 ± 0.11 | 0.80 ± 0.07 | 0.69 ± 0.08 |
| SVM | 0.63 ± 0.11 | 0.73 ± 0.08 | 0.53 ± 0.09 |
| KNN | 0.65 ± 0.11 | 0.77 ± 0.07 | 0.58 ± 0.08 |

AUROC: Area Under Receiver Operating Curve. LR: Logistic Regression. SVM: Support Vector Machine associated with a Gaussian Kernel. KNN: $k$-Nearest Neighbors algorithm, with $k = 10$. AUROC are expressed ± the width of a 95% confidence interval.

## Discussion

This study compared the relevance of several ICP signal descriptors for characterizing ICC in patients with acute brain injury. To the best of our knowledge, this is the first study to compare such ICC indices with the response to a positional shift. We enrolled a cohort of 54 patients with a total of 108 recordings. Most patients underwent multiple ICC tests. The study population encompasses a representative cohort of patients admitted to a neurological intensive care unit with a wide range of ages, pathologies, and initial severity (see Table 1).

ICC test results were used to separate patients according to the induced increase in ICP. Three classes of ICC were defined ("poor", "medium" and "good" ICC) by setting two decision thresholds of 7 mmHg and 10 mmHg. ICP signals were described using a combination of six well-known ICC descriptors and seven EEMD-derived IMFs. These thirteen time series (including the raw ICP signal itself) were used to extract 73 potential ICC surrogates. After a BoLasso-based feature ranking process, three classification models (LR, SVM + Gaussian Kernel, and KNN) were fed with the highest ranked $n$ features, where $n$ ranges from 1 to 10. An input set of $n = 5$ features provided the best compromise between performance, as assessed by computing a cross-validated AUROC, and model complexity (see Fig 7).

### Effects of the positional shift

The positional shift was associated with a significant increase in ICP (mean = 8.7 mmHg, std = 3.4 mmHg, see Table 2). These observations align with previous studies involving comparable maneuvers in acute brain injury patient cohorts [55, 56]. The ICP elevation observed during the maneuver can mostly be attributed to an augmentation in CSF volume within the cranial compartment. Unlike the dura mater around the encephalon, which adheres to the bone, the dura mater at spinal level can distend, especially in a perpendicular direction [57]. Consequently, spinal volume fluctuates in response to changes in position and abdominal pressure. In the supine position, some of the CSF volume contained in the dural sac moves into the cranial compartment. [27]. Furthermore, the total cerebral blood volume increases when the patient is lying down due to vascular redistribution phenomena, especially venous drainage [58]. The present results also suggest further investigation on the relationship between head position and ICP signal morphology. Despite significant ICP elevation in the supine position, AMP remained constant throughout the maneuver in most cases, which is consistent with previous comparable studies [59, 60]. In parallel, the overall ICP pulse morphology remained unchanged, except for the few patients who showed considerable differences between both positions 3. Additional monitoring or imaging would be required to draw confident conclusions about these side effects of the maneuver.

## Selected features

The following feature set was associated with the highest AUROC for all the three classification algorithms: IMF4_hurst, IMF6_amplitude, AMP_median, IMF2_hurst, and IMF2_centroid. Interestingly, this input set is a combination of ICP signal features calculated over various time scales, including single pulse characteristics (AMP and IMF2-derived features), respiratory wave regularity (IMF4_hurst), and slow wave amplitude (IMF6_amplitude). Of those features, only AMP was not derived from the EEMD procedure. AMP is a well-studied ICC surrogate that is sometimes used as a prognostic factor [61]. It is noteworthy that the P2/P1 ratio, classically used to characterise the ICC via the pulse waveform, was not selected in the final classification model. Its absence should not be interpreted as a challenge to its well-established clinical relevance [24, 62]. In fact, this index had a high Spearman correlation with the two selected IMF2-related features, i.e., 0.71 for the Hurst exponent and -0.37 for the frequency centroid. These results suggest that the information provided by the P2/P1 ratio was already embedded in these two ICP signal features.

In any case, the study demonstrated the relevance of EEMD for the delineation of ICP signal components and the establishment of new ICC surrogates. The advantage of using this signal decomposition framework is twofold: First, the iterative nature of EEMD facilitates comparison between IMFs of the same order, regardless of variations in heart rate or respiratory rate, provided the frequency ranges are not too dissimilar. Second, EEMD can handle nonlinear mechanisms as well as nonstationary time series [63], which is valuable when studying ICP signals. As a matter of fact, a few classical ICC surrogates such as RAP and RAQ indices were specifically designed to quantify non-linear phenomena [42].

Notable predictors of patient response were found when examining the respective morphology of IMFs #2, #4 and #6. Over the entire dataset, the average IMF2 frequency centroid was of 2.75 Hz. Therefore, this IMF captures information related to the overall shape of the ICP wave. In particular, its corresponding Hurst exponent was associated with more triangular ICP pulses (see figures in S1 and S2 Figs), which is known to be characteristic of poor ICC [18]. In contrast, IMF4 was associated with frequencies close to those of respiratory waves, with a mean frequency centroid of 0.42 Hz. Its corresponding Hurst exponent can be interpreted as an indicator of respiratory waveform regularity (see figures in S3 and S4 Figs). Interestingly, assisted ventilation appeared to have a negligible effect on IMF4 morphology (see Table 4). As the ICP signal respiratory component is likely due to cerebral venous volume displacements driven by changes in intrathoracic pressure throughout the respiratory cycle [64, 65], a well-defined sinusoid observed in the IMF4 frequency range can be interpreted as impaired compensation mechanisms, just as AMP. In other words, in patients with impaired ICC, cerebral blood volume changes caused by the respiratory cycle are clearly visible in ICP signal, as described by Marmarou's pressure-volume model [13]. Further down in the low frequencies, IMF6 also contains valuable information about the ICC. Over the entire dataset, its average centroid was of 0.06 Hz. ICP signals with higher IMF6 amplitude exhibited clearly visible vasogenic waves that could correspond to B waves (see figures in S5 and S6 Figs), known to be associated with impaired ICC [66]. As originally described by Lundberg *et al.*, [67], lower is the ICC, higher is the amplitude of the vasogenic waves. These slow waves appeared to be clearly attenuated by CSF drainage devices, as seen in Table 5. Overall, each of the three classification models performed best when fed with features related to different time scales. Notably, the mean ICP did not correlate with ICP elevation (Spearman correlation = 0.03). This clearly demonstrates the importance of more advanced ICP signal analysis to get the most out of such an invasive monitoring.

Patient age appeared to have a significant impact on the ICP signal morphology. ICP signals from older patients had a higher pulse amplitude (correlation = 0.34) and av more triangular

pulse shape (assessed with IMF2_hurst, correlation = 0.32). Contrary to expectations, older patients were found to have more pathological ICP pulse morphology, but no worse clinical status or tolerance to supine position. The literature is mixed on the relationship between age and ICC. Although resistance to CSF outflow increases in patients older than 55 years [68], aging does not appear to affect the pressure-volume relationship assessed over short time scales [69], especially in patients who do not suffer from intracranial hypertension [70]. Both conditions (short time scale, no intracranial hypertension) are met by our current protocol. Regarding the pulse morphology, MRI experiments showed that the dampening of pulsatile blood flow in cerebral arteries is altered by aging [71], as arterial stiffness increases [72]. On another note, it is generally admitted that subpeak P2, located at the apex of cardiac-induced triangular ICP pulses, corresponds to a reflection wave that coincides with the maximum volume in the cerebral arteries [37]. It is therefore plausible that reflection waves become stronger with age, leading to more triangular pulse shapes while short-term ICC remains unaltered.

## Limitations

The primary limitation lies in the imperfection of the ICC characterization method. Although the induced increase in ICP depends mainly on the patient's initial ICC, it was not possible to assess the exact CSF and blood volume shifts, specific to each patient's physiology.

Alternative methods have been considered for classifying patients based on ICC, as direct measurements involving CSF addition or withdrawal [73] are not feasible in cohorts of acutely head-injured patients. The change in ICP after jugular compression has been described in the literature as a rapid ICC evaluation [74], but we felt that this would have been a less reproducible protocol in the context of a four-center clinical study. MRI could have also been used to assess ICC, but we would have been limited by the ability of patients to tolerate the supine position for a significant period of time. A major advantage of our approach is its integration into ICU routine practice. Indeed, ICU patients are incidentally placed in the supine position for a few minutes multiple times a day, mainly for nursing or imaging purposes. In the four centers participating in this study, this quick maneuver is even utilized by clinicians to gain insights into patient ICC and to assess the feasibility of conducting imaging or surgery in the supine position.

The second limitation comes from the thresholds used to classify patients according to the results of the ICC test. In this study, we considered that an ICP increase greater than 10 mmHg when moving from a position of 30˚ to 0˚ was a sign of poor ICC. Its counterpart threshold (7 mmHg) was only chosen to be symmetrical with respect to the distribution obtained. The choice of 10 mmHg, however debatable, was defined for several reasons. First, this value was independently suggested by practitioners in the four centers prior to the inclusions. This threshold is also approximately one standard deviation above the average rise in ICP observed in previous studies with similar handling and cohorts [56, 75]. By dividing patients into three classes, the main goal was to quickly identify those patients with the most impaired ICC who might benefit from an adjustment in therapy. Interestingly, our three classification models were better at separating patients above the 10 mmHg-threshold than those below the 7 mmHg threshold (see Table 6), which supports a certain pathological significance of this "bad ICC" threshold.

It is noteworthy that some well-known ICC surrogates were not included in this study. Fourier-based centroids (i.e., Higher Harmonic Centroid and High Frequency Centroid, see [62] for a detailed review) were not included because the underlying physiological mechanisms remain unclear in the context of brain injured cohorts. Because this preliminary study focuses

on the informativeness of the ICP signal alone, indices requiring other signals such as arterial blood pressure, cerebral blood flow or an electrocardiogram were also discarded.

## Conclusion

Univariate ICP signal analysis can be used to discriminate patients with impaired ICC, whereas the mean ICP alone does not provide sufficient information to characterize this mechanical property of the cerebrospinal system. The most accurate statistical models integrate information extracted from the ICP signal at different time scales. From this perspective, EEMD appears to be a valuable tool for isolating multiple components from the raw ICP signal. Further investigation is needed to improve model performance and ultimately integrate ICC characterization into routine bedside monitoring.

## Supporting information

**S1 Fig. Ten ICP signals ordered according to their IMF2 Hurst exponent.** Left column: Five lowest Hurst exponents over the whole dataset. Right column: Five highest Hurst exponents over the whole dataset.
(TIFF)

**S2 Fig. Ten IMF2 ordered according to their Hurst exponent.** Left column: Five lowest Hurst exponents over the whole dataset. Right column: Five highest Hurst exponents over the whole dataset.
(TIFF)

**S3 Fig. Ten ICP signals ordered according to their IMF4 Hurst exponent.** Left column: Five lowest Hurst exponents over the whole dataset. Right column: Five highest Hurst exponents over the whole dataset.
(TIFF)

**S4 Fig. Ten IMF4 ordered according to their Hurst exponent.** Left column: Five lowest Hurst exponents over the whole dataset. Right column: Five highest Hurst exponents over the whole dataset.
(TIFF)

**S5 Fig. Ten ICP signals ordered according to their IMF6 amplitude.** Left column: Five lowest amplitudes over the whole dataset. Right column: Five highest amplitudes over the whole dataset.
(TIFF)

**S6 Fig. Ten IMF6 ordered according to their amplitude.** Left column: Five lowest amplitudes over the whole dataset. Right column: Five highest amplitudes over the whole dataset.
(TIFF)

**S1 Data. Processed time series features.** The 73 first columns correspond to the time series features considered in the study. Patient responses (i.e., ICP elevation) can be calculated by subtracting the column "BASAL_ICP" to the column "PLATEAU_ICP".
(CSV)

## Acknowledgments

The authors would like to thank Dr. Julien Henriet and Dr. Jean-Christophe Lapayre for their guidance and support throughout the project.

## Author Contributions

**Conceptualization:** Jean-François Payen, Yoann Launey, Laurent Gergelé.

**Data curation:** Donatien Legé, Pierre-Henri Murgat, Russell Chabanne, Kevin Lagarde, Yoann Launey, Laurent Gergelé.

**Formal analysis:** Donatien Legé.

**Funding acquisition:** Marion Prud'homme, Laurent Gergelé.

**Investigation:** Donatien Legé, Pierre-Henri Murgat, Russell Chabanne, Kevin Lagarde, Clément Magand, Yoann Launey, Laurent Gergelé.

**Methodology:** Donatien Legé, Jean-François Payen.

**Project administration:** Marion Prud'homme, Laurent Gergelé.

**Resources:** Marion Prud'homme.

**Software:** Donatien Legé.

**Supervision:** Marion Prud'homme.

**Validation:** Donatien Legé, Yoann Launey.

**Visualization:** Donatien Legé.

**Writing – original draft:** Donatien Legé, Pierre-Henri Murgat.

**Writing – review & editing:** Donatien Legé, Marion Prud'homme, Yoann Launey, Laurent Gergelé.

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
