## [Decision Letter · Decision Letter 0]

13 Nov 2024

PONE-D-24-37894Cerebral compliance assessment from intracranial pressure waveform analysis: Is a positional shift-related increase in intracranial pressure predictable?PLOS ONE

Dear Dr. Legé,

Thank you for submitting your manuscript to PLOS ONE. After careful consideration, we feel that it has merit but does not fully meet PLOS ONE’s publication criteria as it currently stands. Therefore, we invite you to submit a revised version of the manuscript that addresses the points raised during the review process.

We look forward to receiving your revised manuscript.

Kind regards,

Yasin Hamarat

Academic Editor

PLOS ONE

Journal Requirements: When submitting your revision, we need you to address these additional requirements. 1. Please ensure that your manuscript meets PLOS ONE's style requirements, including those for file naming. The PLOS ONE style templates can be found at https://journals.plos.org/plosone/s/file?id=wjVg/PLOSOne_formatting_sample_main_body.pdf and https://journals.plos.org/plosone/s/file?id=ba62/PLOSOne_formatting_sample_title_authors_affiliations.pdf 2. Thank you for stating the following in the Competing Interests section: "Marion Prud'homme and Donatien Legé are employees of Sophysa Company. Laurent Gergelé has performedconsulting work for Sophysa Company."  Please confirm that this does not alter your adherence to all PLOS ONE policies on sharing data and materials, by including the following statement: ""This does not alter our adherence to  PLOS ONE policies on sharing data and materials.” (as detailed online in our guide for authors http://journals.plos.org/plosone/s/competing-interests).  If there are restrictions on sharing of data and/or materials, please state these. Please note that we cannot proceed with consideration of your article until this information has been declared.  Please include your updated Competing Interests statement in your cover letter; we will change the online submission form on your behalf. 3. Please review your reference list to ensure that it is complete and correct. If you have cited papers that have been retracted, please include the rationale for doing so in the manuscript text, or remove these references and replace them with relevant current references. Any changes to the reference list should be mentioned in the rebuttal letter that accompanies your revised manuscript. If you need to cite a retracted article, indicate the article’s retracted status in the References list and also include a citation and full reference for the retraction notice.

**Additional Editor Comments:**

Thank you for submitting your manuscript. I found it both informative and well-written, offering valuable insights into cerebrospinal compliance assessment. Your results indicate that most patients had stable compliance, as shown by the flat zone of the intracranial pressure-volume curve. It would be helpful to add a brief comment to make it clear that this doesn’t mean the P1/P2 ratio isn’t useful—it still helps track relative changes in cerebrospinal compliance. This will prevent readers from getting the wrong impression. Please take a look at the attached reviewer comments for some suggested clarifications to strengthen your work even further. Minor language editing is needed to ensure the manuscript is at its best. 

Reviewers' comments:

Reviewer's Responses to Questions

**Comments to the Author**

1. Is the manuscript technically sound, and do the data support the conclusions?

Reviewer #1: Yes

Reviewer #2: Yes

2. Has the statistical analysis been performed appropriately and rigorously? 

Reviewer #1: Yes

Reviewer #2: Yes

3. Have the authors made all data underlying the findings in their manuscript fully available?

Reviewer #1: Yes

Reviewer #2: Yes

4. Is the manuscript presented in an intelligible fashion and written in standard English?

Reviewer #1: Yes

Reviewer #2: Yes

5. Review Comments to the Author

Reviewer #1: 1. Congratulations. Perfect science.

2. Please, include information on patients' outcomes into a revised manuscript.

3. Regarding a discussion. Your results show that just a few patients had a low compliance. All others were in flat zone of intracranial pressure -volume curve. Please, include comment on that in order not to create a wrong impression of paper's readers that P2/P1 ratio is not associated with intracraniospinal compliance changes.

Reviewer #2: Well written paper, relevant topic. In this paper, 73 ICP signal features were calculated over the 20

minutes prior to the ICC test. After a selection step, different combinations of these

features were provided as inputs to classification models. The goal was to predict the

level of induced ICP elevation, which was categorized into three classes: less than 7

mmHg ("good ICC"), between 7 and 10 mmHg ("medium ICC"), and more than 10

mmHg ("poor ICC"). A logistic regression model fed with a combination of 5 ICP signal

features discriminated the "poor ICC" class with an area under the receiving operator

curve (AUROC) of 0.80 (95\\%-CI : [0.73 - 0.87]). The overall one-versus-one

classification task was achieved with an averaged AUROC of 0.72 (95\\%-CI : [0.61 -

0.83]).

6. PLOS authors have the option to publish the peer review history of their article (what does this mean?). If published, this will include your full peer review and any attached files.

Reviewer #1: No

Reviewer #2: No

---

## [Author Response · Author response to Decision Letter 0]

5 Dec 2024

Editor: It would be helpful to add a brief comment to make it clear that this doesn’t mean the P1/P2 ratio isn’t useful—it still helps track relative changes in cerebrospinal compliance. This will prevent readers from getting the wrong impression. 

Reviewer 1: Please, include comment on that in order not to create a wrong impression of paper's readers that P2/P1 ratio is not associated with intracraniospinal compliance changes.

We integrated a paragraph in the discussion to precise our findings concerning the P2/P1 ratio (lines 413 – 419). As explicitely stated, our results do not call into question the relevance of the P2/P1 ratio.

-

Reviewer 1: Please, include information on patients' outcomes into a revised manuscript.

We included in the cohort description (Table 1) the 28-day mortality as well as the length of stay in an Intensive Care Unit.

---

## [Editor Report · Decision Letter 1]

9 Dec 2024

Cerebral compliance assessment from intracranial pressure waveform analysis: Is a positional shift-related increase in intracranial pressure predictable?

PONE-D-24-37894R1

Dear Dr. Legé,

We’re pleased to inform you that your manuscript has been judged scientifically suitable for publication and will be formally accepted for publication once it meets all outstanding technical requirements.

Kind regards,

Yasin Hamarat

Academic Editor

PLOS ONE

---

## [Editor Report · Acceptance letter]

13 Dec 2024

PONE-D-24-37894R1 

PLOS ONE

Dear Dr. Legé, 

I'm pleased to inform you that your manuscript has been deemed suitable for publication in PLOS ONE. Congratulations! Your manuscript is now being handed over to our production team.

Kind regards, 

on behalf of

Dr. Yasin Hamarat 

Academic Editor

PLOS ONE